# A Portable Multi-Modal Cushion for Continuous Monitoring of a Driver’s Vital Signs

**DOI:** 10.3390/s23084002

**Published:** 2023-04-14

**Authors:** Onno Linschmann, Durmus Umutcan Uguz, Bianca Romanski, Immo Baarlink, Pujitha Gunaratne, Steffen Leonhardt, Marian Walter, Markus Lueken

**Affiliations:** 1Medical Information Technology, Helmholtz Institute, RWTH Aachen University, 52074 Aachen, Germany; 2Toyota Collaborative Safety Research Center, Toyota Motors Corporation, Ann Arbor, MI 48105, USA

**Keywords:** unobtrusive sensing, capacitive ECG, magnetic induction monitoring, reflective PPG, seismocardiography, private space, automotive

## Abstract

With higher levels of automation in vehicles, the need for robust driver monitoring systems increases, since it must be ensured that the driver can intervene at any moment. Drowsiness, stress and alcohol are still the main sources of driver distraction. However, physiological problems such as heart attacks and strokes also exhibit a significant risk for driver safety, especially with respect to the ageing population. In this paper, a portable cushion with four sensor units with multiple measurement modalities is presented. Capacitive electrocardiography, reflective photophlethysmography, magnetic induction measurement and seismocardiography are performed with the embedded sensors. The device can monitor the heart and respiratory rates of a vehicle driver. The promising results of the first proof-of-concept study with twenty participants in a driving simulator not only demonstrate the accuracy of the heart (above 70% of medical-grade heart rate estimations according to IEC 60601-2-27) and respiratory rate measurements (around 30% with errors below 2 BPM), but also that the cushion might be useful to monitor morphological changes in the capacitive electrocardiogram in some cases. The measurements can potentially be used to detect drowsiness and stress and thus the fitness of the driver, since heart rate variability and breathing rate variability can be captured. They are also useful for the early prediction of cardiovascular diseases, one of the main reasons for premature death. The data are publicly available in the UnoVis dataset.

## 1. Introduction

Many traffic accidents can be traced back to drowsiness, stress and other serious physiological states, such as heart attacks and strokes [1,2,3]. In ageing societies, it can be expected that the number of car accidents related to physiological problems will increase. In partly autonomous vehicles, driver monitoring systems are crucial to ensure that the vehicle driver can take over control at any moment [4]. Personal healthcare systems, such as in-vehicle monitoring, increase the coverage of health data. Not only with respect to the elderly, they support the early detection of cardiovascular diseases, one of the leading causes of premature death [5]. For the assessment of health status, important vital signs are the heart rate (HR) and respiratory rate (RR). Changes with respect to the HR can not only give insights into (unknown) arrhythmias, such as atrial fibrillation, but also into the state of the central nervous system via heart rate variability parameters (HRV) [6]. HRV can be used to assess, for example, drowsiness and stress [7,8,9]. RR, and especially its variability, may give further insights into drug abuse [10] and major cardiac events, even before any changes in HRV are noticeable [11,12].

Various systems for the in-vehicle monitoring of vital signs exist. These include optical sensors (reflective photoplethysmography, wrist-worn devices [13]), camera systems (RGB camera [14], infra-red thermography camera [15]), radar [16] and capacitive electrocardiography [17]. Using these sensors, the driver’s physiological status can be extracted by means of their heart rate, respiratory rate, heart rate variability (HRV) and oxygen saturation [18,19]. These values can be analysed to detect stress, drowsiness, heart attacks, strokes or atrial fibrillation [20,21,22]. If monitored continuously over time, trends with respect to the overall health status and the early detection of physiological complications can be assessed. However, in contrast to conventional contact-based methods, unobtrusive sensing methods suffer from motion artifacts, since the coupling with the user is not fixed and is unknown beforehand. RGB camera-based methods may additionally suffer from changes in illumination.

Existing systems integrating several unobtrusive modalities into a car include the so-called U-car presented by Baek et al. [23], a system presented by Warnecke et al. [24] and a system presented by Leicht et al. [25]. The system of Baek et al. [23] consists of a one-lead capacitive electrocardiography (cECG) sensor in the backrest of the seat, with ECG, galvanic skin response and reflective photoplethysmography (rPPG) sensors embedded into the steering wheel and a piezoelectric sensor in the seat belt. The system of Warnecke et al. [24] consists of a ballistocardiography sensor in the backrest of the seat, an ECG and reflective photoplethysmography sensors in the steering wheel and a camera for photoplethysmography imaging (often also called remote PPG, imaging PPG or camera-based PPG). Leicht et al. [25] embedded a six-lead capacitive ECG system into the backrest of the seat, a sensor consisting of magnetic induction measurement, reflective PPG and cECG in the seat belt and an RGB and infrared thermography camera in the front panel of the car. A comprehensive list of monitoring systems used in cars, including systems using only one modality, can be found in the reviews of Leonhardt et al. [18] and Wang et al. [26]. Monitoring systems embedded into the steering wheel heavily depend on the continuous contact of the hands at specific locations on the steering wheel. Camera-based systems are often accompanied by privacy concerns for the vehicle driver and require dedicated systems to process the video data. Systems embedded into the seat belt and the driver seat can lead to certification issues [25]. Furthermore, most of the systems in the literature only provide one channel of each modality, which could lead to a complete loss of signal if the sensor contact is disturbed. Finally, publicly available datasets for unobtrusive modalities are sparse [27], especially with respect to driving scenarios. Thus, the comparability of approaches for motion artifact compensation, signal quality assessment and diagnostic applications (drowsiness detection, stress) is limited. For completeness, it should be mentioned that multi-modal unobtrusive sensors are also used in other applications and embedded into other everyday objects, such as armchairs [28], office chairs [29] and mattresses [30].

In this paper, a portable cushion is presented in which four measuring modalities for monitoring the most important vital signs, i.e., heart rate and respiratory rate, are embedded. The four modalities include capacitive electrocardiography (cECG), reflective photoplethysmography (rPPG), magnetic induction measurement (MIM) and seismocardiography (SCG). They are integrated into four redundant sensor units, called 4xU sensors, to increase coverage with respect to different seating positions and statures. Provided an accordingly high SNR, the cECG may allow an even more comprehensive diagnosis related to cardiovascular diseases than by merely monitoring the heart rate. By introducing redundancy, the loss of coverage due to motion artifacts and other changes with respect to the coupling of the body and sensor may be reduced. Furthermore, redundancy enables the use of sensor fusion algorithms. Advantages over camera systems and sensors directly embedded in the driver’s seat include privacy concerns and certification issues. With respect to systems embedded into the steering wheel, the system does not rely on specific contact with the driver other than that the driver is seated. Additionally, the cushion can be placed on other seats, such as armchairs, sofas or office chairs, making it a universal monitoring tool.

The system was tested with 20 subjects driving in a driving simulator for around 25 min. The measurements were compared with the gold standard, the conductive ECG and impedance pneumography. The data recorded from the cushion and the reference monitor have been made publicly available in the UnoVis dataset to increase the comparability of the algorithmic approaches of different research groups (https://www.medit.hia.rwth-aachen.de/en/publications/unovis (accessed on 27 February 2023)). To the best of the authors’ knowledge, this is the first publicly available dataset of multi-modal unobtrusive sensors in a driving scenario. The recorded camera data are available upon reasonable request. The paper’s main contributions are as follows:An integrated, portable device with dedicated hardware for measuring HR and RR unobtrusively.A detailed technological description of the integration with respect to all components.A publicly available dataset of the recorded sensor data in a real-world scenario without constraints on clothes and movement. The data should enable researchers to verify the presented results, test their own sensor fusion algorithms and contribute to the training of machine learning models. Furthermore, the data should contribute to the data-sharing paradigm.An analysis of the vital signs showing high coverage and high quality for HR and fair quality for RR with respect to each modality independently. The analysis should give new insights into problems and opportunities for each modality with respect to different driving situations.

The rest of the paper is structured as follows. First, the theoretical background of the measurement modalities used in the proposed system is presented. Second, the complete system setup is explained in detail. Third, the preliminary results of the first proof-of-concept study with 20 participants are analysed. Finally, the results are summarised and discussed.

## 2. Materials and Methods

Several approaches exist to measure vital signs unobtrusively. In the following, the technical principles of the cECG, the reflective PPG, the MIM and the SCG are presented. The cECG, reflective PPG and SCG can be applied with the aim to extract the heart rate and HRV. However, the respiratory rate can often be derived in addition. For determining the respiratory rate, MIM may be employed.

### 2.1. Capacitive Electrocardiography

Due to the electrical activation of the heart muscle during each heart beat, heart activity can be measured by means of a differential (biopotential) measurement, called electrocardiography. The ECG is the gold standard for diagnosis in cardiology [31]. Traditionally, the ECG is measured using adhesive electrodes, which are attached directly onto the skin. As early as in 1967, the use of dry electrodes was introduced by Richardson [32], i.e., conductive plates (e.g., copper or tin) for measuring ECG unobtrusively. This form of ECG enables a measurement without adhesive electrodes and without direct skin contact (e.g., through clothes) and thus capacitive measurement in private spaces.

Since there is no direct conductive contact with the skin, there is high coupling impedance between the skin and sensor surface. Hence, a trans-impedance amplifier with a very high input impedance is necessary to measure the skin potential leading to an “active electrode” [18]. A simplified model of the electrode principle is depicted in Figure 1. The skin potential is modelled by the input voltage source, and the skin–electrode path is modelled by a high coupling resistor $R_{c}$ and a coupling capacitor $C_{c}$. $C_{in}$ and $R_{in}$ describe the amplifier’s input impedance. On the one hand, $R_{in}$ has to be very high (gigaohms) to reduce the high-pass effect due to $C_{c}$. On the other hand, triboelectricities on the capacitor need a long time to discharge due to the high value of $R_{in}$. Therefore, the cECG is more sensitive to noise and more affected by motion artifacts than the conductive one.

Finally, the ECG is obtained by measuring the differential signals of two electrodes, the so-called “leads”. Power-line noise and other common mode disturbances make it necessary to use differential amplifiers with a high common-mode rejection ratio. Additionally, the so-called driven right leg (DRL) circuit can be used to further suppress common-mode noise by means of an additional electrode [33]. For the cECG, this is usually realised using a conductive plate, on which the subject sits. Similarly to the conductive case, the sum signal of each active electrode’s output is fed back negatively onto the conductive plate, which again has capacitive coupling with the skin. In contrast to the active electrodes used for obtaining the leads, no active electrode is used for the DRL circuit [34].

### 2.2. Reflective Photoplethysmography

With every heart beat, blood is pumped through the vessels into the body. This change in blood volume can be measured by means of an optical system, called photoplethysmography (PPG). PPG can either be measured transmissively or reflectively. For the reflective method shown in Figure 2, light is emitted into the body’s tissue by means of an LED. At the same time, the reflected light is measured with a photodiode (PD) placed close to the LED. The amount of the reflected light is modulated by the blood volume and thus changes with the pulse. If the illumination strength and wavelength of the LED are chosen appropriately (e.g., infrared light), reflective PPG can also be measured through layers of clothing. This is possible when several LEDs are placed around a photodiode to increase the illumination of the tissue and thus the diffuse reflection. Here, the clothing can be assumed to introduce another damping layer [7,35]. Even if the tissue is not reached, the deformation of the layer of clothes due to the heart’s response (optical ballistocardiography) can often still be measured [36].

### 2.3. Magnetic Induction Measurement

Magnetic induction measurement is an unobtrusive method to measure the respiratory rate. Its principle was first introduced in 1967 by Vas et al. [38]. In the literature, both gardiometer-based and oscillator-based approaches have been introduced. For the latter, the idea is to have an electric oscillator (e.g., Colpitts oscillator) in which a planar coil is the frequency-determining component. Due to the oscillating current through the coil, a primary magnetic field is generated (Biot–Savart law). The coil is placed in such a way that its primary field induces an induction voltage and thus an eddy current in the tissue of a subject (cf. Figure 3). This eddy current again generates a secondary magnetic field that affects the primary one, modulating the oscillator’s frequency depending on the tissue’s impedance. If the coil is placed close to the lungs, the eddy current changes with the lung volume and thus the respiration can be measured by recording the change in frequency of the oscillator.

### 2.4. Seismocardiography

Seismocardiography refers to the measurement of the mechanical vibrations of the heart beat on the body’s surface by means of an accelerometer [39]. For this, a simple micro-electrical-mechanical sensor can be placed, for example, on the thorax of a subject. Each heart beat produces a distinct waveform, where the different peaks coincide with different phases of the heart cycle (cf. [39] for more information). Additionally, the respiratory movement can be obtained.

## 3. System Setup

The system embedded into the cushion relates to the system of Yu et al. [30] and consists of two elements (cf. Figure 4). First, it consists of four 4xU sensor units that include a reflective PPG sensing unit, an MIM sensing unit, an accelerometer and one active electrode for cECG (top right side). Each of the four sensors is indicated by a red number in Figure 4. The reflective PPG, MIM and SCG sensing units all provide digital signals, whereas the active electrodes for the cECG provide an analogue signal. Second, it is composed of the so-called controller box, which reads the measurements of the 4xU sensors, captures the cECG signals and provides a user interface to access the data.

### 3.1. 4xU Sensors

Each 4xU sensor unit consists of one active electrode (analogue), a unit for reflective PPG measurements, a unit for SCG measurements and MIM (digital). All digital signals are processed by an STM32 microcontroller (μC), STM32F303CB6T (STMicroelectronics N.V., Plan-les-Quates, Switzerland). The sensor’s front side is shown in Figure 4. Each 4xU sensor’s digital signals are processed by a dedicated μC, firstly to introduce an abstraction layer with a communication protocol to ease upgradability, secondly to reduce the computational workload of the μC in the controller box and thirdly to isolate the sensor readings, since the analogue front ends used on each 4xU sensor have the same addresses on the bus.

The cECG electrode is realised using a shielded high-impedance operational amplifier, OPA140 (Texas Instruments, Dallas, TX, USA), with an input impedance of 10 TΩ and 10 pF. The output of the amplifier is led to the controller box using a coaxial cable, where the ECG leads are obtained. A conductive fibre is sewn onto the seating area and used as the driven right leg circuit’s electrode.

The reflective PPG measurement is controlled by an analogue front-end, ADPD1080 (Analog Devices, Norwood, MA, USA), which is accessed by the microcontroller via an I2C bus. For the reflective PPG measurement, three infrared (IR) LEDs, SFH4250 (Osram Licht AG, Munich, Germany), and one IR-sensitive photodiode, BPW34 (Osram Licht AG, Munich, Germany), are used. The LEDs have a distance of 2 cm from the photodiode and are placed at 30, 150 and 270 degrees. The ADPD1080 is programmed in such a way that the LEDs are pulsed with 3 μs pulse width and six pulses per measurement. Simultaneously, the photodiode is read such that ambient light is suppressed and the signal-to-noise ratio (SNR) is improved. Ambient light suppression is achieved by subtracting the measured voltage when the LED is not active and thus only measuring the ambient light.

The MIM is realised using a Colpitts oscillator with a planar coil that has an inner diameter of 78 mm and an outer diameter of 88 mm. The coil has five windings. The frequency of the Colpitts oscillator is measured by means of a dual inverter, SN74LC2G04 (Texas Instruments, Dallas, TX, USA), whose output’s frequency is measured by the counter input of the μC. The frequency of the oscillator can be varied since the circuit includes two varicaps, BBY66 (Infineon Technologies AG, Neubiberg, Germany), which act as voltage-dependent capacitors in such a way that their value can be adjusted using the analogue output of the μC. The fine adjustment of the frequency is necessary to suppress crosstalk between the four sensors [40]. The frequency is around 100 MHz. According to [40], the sensors are placed more than 10 cm apart from each other to reduce crosstalk.

The seismocardiography measurement is performed by the BMI160 inertial measurement unit (Bosch Sensortec GmbH, Reutlingen, Germany). The BMI160 provides a 16-bit, three-axis measurement of an accelerometer. Furthermore, it provides a gyroscope, which is not used in the presented setup. The sensor is placed on the bottom side of the PCB and connected to the μC using the I2C bus. The *z*-axis of the BMI points to the inside of the cushion.

The sensor channels (except cECG) have been enumerated according to the sensor nodes from the top left to the bottom right, from right to left and top to bottom (cf. Figure 4). The cECG is enumerated according to the leads obtained (see below).

### 3.2. Controller Box

The controller box consists of a microcontroller board, STMF303-Nucleo (STMicroelectronics N.V., Plan-les-Quates, Switzerland), an analogue front end for calculating the cECG leads, ADS1298 (Texas Instruments, Dallas, TX, USA), and a real-time clock (RTC), DS3231 (Analog Devices, Norwood, MA, USA). Additional to the USB micro output of the Nucleo board, an SD card is used to store the raw data. The controller box can be supplied by either a 12 V battery, Makita 197396-9 (Makita, Anjo, Aichi, Japan), or by any 12 V medical-grade power adapter. The 12 V supply is converted into ±5 V for the symmetric supply of the OPA140 on each sensor using a Traco DC-DC converter, TMV 1205 (Traco Electronic AG, Baar, Switzerland).

The output of each OPA140 is connected to the ADS1298 using a coaxial cable. The differential signals between the upper left and right (lead one), the upper left and lower right (lead two), the upper right and lower left (lead three) and the lower left and right (lead four) are calculated. The first two leads correspond to the Einthoven I and II leads [31]. The ADS1298 provides a high common-mode rejection ratio to suppress power-line noise and a programmable driven right leg output. Additionally, it provides a 24-bit sigma-delta analogue-to-digital converter. The digitalised signal is accessed by the Nucleo board via an SPI bus.

All four sensors are connected with the controller box board using an RS485 interface and one UART output interface of the microcontroller. The protocol for reading the sensors is a two-step process. First, a read command for all sensor units is sent. Second, each sensor unit’s values are read using the sensor’s identifier. By this procedure, a synchronised reading of the reflective PPG, MIM and SCG is ensured. The sampling frequency for each modality is 128 Hz. The cECG is sampled at the same time that the read command is sent.

The data of the system can be either accessed via the USB micro port of the Nucleo board by a virtual COM port or by an SD card on which the data are stored as CSV files. The real-time clock of the system is used to create time stamps and can be set by a software program, if the system is connected via USB.

## 4. Experimental Evaluation

To validate the sensor cushion’s functionality, a study with 20 healthy participants was conducted in the driving simulator shown in Figure 5. The open-source simulator CARLA was used for simulating a driving environment [41]. A Logitech driving force GT steering wheel and gas pedals were used for driving. For the simulation, the environment “Town04” was chosen and a custom user interface was created showing the speed and both side mirrors [41]. The participants (2 female, 18 male) were aged between 19 and 60 years old (mean: 25.9 years, SD: 8.5280 years) and had a body mass index between 20.23 kg/$m^{2}$ and 28.7 kg/$m^{2}$ (mean: 23.46 kg/$m^{2}$, SD: 2.44 kg/$m^{2}$). They were mostly wearing one layer of clothing made from cotton. A complete list of the demographics is shown in Table 1.

All participants gave written consent and the study protocol was reviewed by the ethics committee of RWTH Aachen University Hospital (EK 183/22). After the introduction and the provision of written consent, a short test drive of 2 min was performed to rule out any simulator sickness among the participants. Then, the experiment started and was divided into four stages:Driving without talking to simulate a single driver.Controlled movements, which could be expected during driving, i.e., head torsion left/right, body rotation left/right, adjusting the position on the seat, leaning forward. After each movement, a pause of around 10 s was made.Driving while talking to the study staff to simulate a passenger.Sitting in the seat without driving or talking to obtain a clean signal for reference.

The complete protocol is visualised in Figure 5.

Additionally to the signals of the sensor cushion, a video with a bimodal camera capturing an RGB and infrared frame (Optris PI230, Optris GmbH, Berlin, Germany) and a conductive reference ECG measurement with a Philips MX700 patient monitor (Philips, Amsterdam, The Netherlands) were recorded. The patient monitor also provided a respiratory reference by impedance pneumography. The data of the patient monitor were accessed using the software iXtrend 2.1. From the simulation software, the steering angle and the throttle were recorded. The participants were allowed to drive freely without the necessity to obey traffic rules. After the recording, the data from the sensor cushion (RTC time stamp) and the data from the other modalities (PC time stamp) were synchronised manually. The data of the cushion, simulator and patient monitor have been added into the UnoVis dataset (https://www.medit.hia.rwth-aachen.de/en/publications/unovis (accessed on 27 February 2023)). The corresponding camera data are available upon reasonable request.

### 4.1. Preprocessing

All signal modalities recorded may consist of both cardiac-related and respiration-related signals. While the cECG, reflective PPG and SCG are considered cardiac signals, they all exhibit baseline wanders due to the respiration. For the cECG, this is due to changes in the distance and thus impedance changes between two electrodes for one lead during each breathing cycle by the movement of the thorax. For reflective PPG, the baseline wander is due to blood pressure changes due to the respiration and, in the presented setup, also due to compression of the clothing by the movement of the thorax. The SCG captures the movement of the thorax due to the change in its acceleration. The MIM signal is considered mainly breathing-related. However, it was shown to also be able to capture cardiac-related signals if placed close to the heart [36]. In the presented setup, this is not the case.

Each raw signal was filtered to remove noise when analysing the data. All signals consisting of a cardiac and respiratory component were filtered separately by two different digital filters to separate the cardiac signal and the respiration signal (cf. Figure 6).

For the cardiac part of the cECG channels, an 8th-order Butterworth band-pass filter with cut-off frequencies at 0.4 Hz and 45 Hz was used to remove high-frequency noise. Furthermore, a 50 Hz notch filter was applied to remove power-line noise. To extract the respiratory component, the unprocessed signal was also filtered by a 2nd-order Butterworth filter with cut-off frequencies at 0.15 Hz and 0.5 Hz.

The reflective PPG signals were filtered by a 2nd-order Butterworth band-pass filter with cut-off frequencies at 0.5 Hz and 2 Hz to remove noise and extract the cardiac signal. To extract the respiratory component, the unprocessed signals were filtered with a 2nd-order Butterworth filter with cut-off frequencies of 0.1 Hz and 0.4 Hz.

The MIM signals were filtered with a 2nd-order Butterworth band-pass filter with cut-off frequencies at 0.15 Hz and 0.4 Hz to remove noise. Additionally, a tenth-order median filter was applied to remove artifacts.

The SCG signals were filtered with an 8th-order Butterworth band-pass filter with cut-off frequencies at 0.4 Hz and 45 Hz to extract the cardiac signal. To extract the respiratory signal, the unprocessed signal was filtered with a 2nd-order Butterworth band-pass filter with cut-off frequencies at 0.1 Hz and 0.4 Hz.

### 4.2. Extraction of Vital Signs

For the cardiac signals, different standard algorithms for the extraction of the inter-beat intervals (IBI) were applied. First, the Pan–Tompkins algorithm [42] was employed for the cECG and reference ECG; second, a simple peak detection (MATLAB’s *findpeaks*-function) was used for the reflective PPG with a minimum peak distance of 0.5 s; third, the interval estimator according to Brüser et al. [43] was applied for the SCG. The algorithm from Brüser et al. combines a modified autocorrelation, a modified average magnitude difference function and maximum peak pairs to estimate the most likely inter-beat interval for a time window that is chosen in such a way that at least two heart beats lie within it [43]. The heart rate can then be computed from the IBIs by
(1)$${\mathrm{HR}}_{i}=\frac{1}{{\mathrm{IBI}}_{i+1}-{\mathrm{IBI}}_{i}}.$$

For the respiratory signals, i.e., the reference respiration, MIM, reflective PPG-derived respiration, cECG-derived respiration and SCG-derived respiration, simple peak detection (MATLAB’s *findpeaks*-function) algorithms were applied with a minimum peak distance of 2 s. Again, the RR can be computed using the inter-breath interval (IRI):(2)$${\mathrm{RR}}_{i}=\frac{1}{{\mathrm{IRI}}_{i+1}-{\mathrm{IRI}}_{i}}.$$

It should be noted that no algorithms for the removal or compensation of motion artifacts were applied. Furthermore, the signals were analysed independently and no signal fusion was applied to improve estimations of the heart rate or respiratory rate at this point in the project as the aim was to present the technology and provide a publicly available dataset.

## 5. Results

The recorded signals were compared qualitatively and quantitatively. For the quantitative evaluation and comparison with the reference, the so-called success rate described in [37], the coverage with respect to IEC 60601-2-27 according to [44,45] and the Pearson correlation coefficient for the covered segments were used. The success rate gives the percentage of the recording in which the absolute difference between the estimated and reference heart/respiratory rate are bounded by a limit *l*. *l* was varied between 0 BPM and 10 BPM with a step size of 0.1 BPM. Finally, the Area under Curve (AUC) normalised by the total area was used as an evaluation metric. The coverage with respect to IEC 60601-2-27 is defined by the ratio of the estimated heart rate signal fulfilling the accuracy defined in IEC 60601-2-27 to the complete signal. IEC 60601-2-27 requires the estimated heart rate to fulfill
(3)$$|{\mathrm{HR}}_{\mathrm{est}}-{\mathrm{HR}}_{ref}|\leq \max\left(5\;\mathrm{BPM},0.1\;\cdot \;{\mathrm{HR}}_{ref}\right)$$
where ${\mathrm{HR}}_{\mathrm{est}}$ and ${\mathrm{HR}}_{ref}$ are the estimated and reference heart rates, respectively [45]. For the RR, 2 BPM were used instead of 5 BPM. Finally, the Pearson correlation for the segments fulfilling the coverage criterion was computed [46]. Signals with high correlation coefficients are assumed to capture heart rate variability (HRV) or breathing rate variability (BRV) parameters, since the product of the standard deviations of the two tested signals (which is the variability in HR or RR) should be equal to the covariance. The recordings were evaluated with respect to the different driving stages as more motion usually distorts the signals more strongly.

### 5.1. Qualitative Results

During each recording and between individuals, varying signal qualities with respect to the waveform could be observed. The quality of each signal was inspected visually by means of a waveform-related criterion. Manual annotations of the signal quality are available for each subject and each channel. For the annotations, the software Signalplant [47] was used. For the cECG, good quality means that characteristic features of the ECG waveform, i.e., P-wave, QRS complex and T-wave, were visible. In case of good quality, a noise floor was allowed that only obscured the P-wave. Fair quality was obtained when all R-peaks were visible but the signal was distorted by baseline wander and a higher noise floor. Bad quality was obtained when R-peaks were completely obscured by noise, motion artifacts were present or the R-peaks were not clearly distinguishable from noise. For the reflective PPG, good quality means that distinct peaks with each heart beat were visible. Fair quality was noted when each heart beat was visible but the noise floor was high. Bad quality was noted in the event that some heart beats were not visible or motion artifacts were present. For the MIM, good quality means that the change in the frequency of the coil was in phase with the respiratory movement. Fair quality was noted when the respiratory movement was visible but spikes distorted the signal. The quality was assessed as bad if no respiratory signal was visible or the signal was completely distorted by spikes or motion artifacts. For the SCG, good quality means that distinct peaks with each heart beat were visible. In contrast, bad quality was the absence of clear signals related to the heart or respiratory rate. Examples of good and bad quality are depicted in Figure 7 and Figure 8.

The cECGs’ signal quality is not only dependent on the mechanical contact but also on triboelectricities, which need to be discharged, especially in the beginning of the recording. During this time (up to 10 min), the power-line noise decreases continuously. Apart from the typical power-line noise, a noise signal at around 11.7 Hz could be observed. As visible in the first plots of Figure 7 and Figure 8, the cECG may even have waveforms close to the conductive ECG. However, often, only the R-peaks and T-wave were visible. Usually, the first lead and fourth lead of the cECG signals show the best quality. With respect to the participants’ gender, it could be observed that the cECG signals were less reliable and more noisy for females, supposedly because of their longer hair, which introduced more triboelectricities.

The reflective PPG signals do not describe the usual waveform of a clinical reflective PPG but are rather sinus-shaped for each heart beat and superimposed with a sinus coinciding with the respiratory movement. For good-quality reflective PPG (top right plot of Figure 7), each heart beat is described by a distinct peak. In case of bad-quality reflective PPGs (top right plot of Figure 8), the respiratory movement was often still visible.

The MIMs describe sinus-shaped waveforms that were disturbed by spikes (cf. bottom left of Figure 7 and Figure 8). In case of good-quality MIMs, the spikes were sparse and clear respiratory movement was visible. In case of bad-quality MIMs, the spikes were distorting the signals irreversibly such that the respiratory movement could not be extracted.

The SCG signals’ quality is only good for segments with barely any motion. A typical waveform with varying amplitudes superimposed with the respiratory movement is visible on the bottom right of Figure 7. The signals were usually best in the z and x directions of the accelerometer, supposedly because the y axis is pointing in the lateral direction, where the vibration is weakest.

Motion artifacts were present during all stages of the recording. However, especially in the second stage, it could be observed that motion artifacts did not necessarily occur in all modalities or all channels during specific movements (e.g., head torsion). Furthermore, while motion artifacts generated baseline wander with high amplitudes in some cECG recordings, the R-peaks were still visible.

During visual inspection of the respiratory reference, it was observed that, in some cases, the reference signal reached saturation or lost the signal (cf. Figure 9). Therefore, instead of the raw signal of the impedance pneumography, the derived numerical values (1 min^−1^) provided by the patient monitor were used for the quantitative analysis.

### 5.2. Quantitative Results

#### 5.2.1. cECG

From Figure 10, it can be seen that the cECG signals provide the most accurate estimation of the heart rate with respect to the AUC for most stages. Only in stage two, the reflective PPG is more accurate. It can also be seen that the cECG has the highest variance with respect to the AUC in the first stage. This might be due to the discharging process of triboelectricites. In the motionless stages, cECG leads one and four were the most accurate. In the other stages, either lead one (stage three) or lead four (stage two) was the most accurate. Lead three was always the most inaccurate. A median coverage above 74% could be achieved for channels one, two and four (cf. Table 2). Channel three only had a median coverage of 34%. The highest achieved coverage was 95.9% for participant three. With respect to these segments, Pearson correlation coefficients of above 0.95 (*p*-value < 0.05) could be achieved. With respect to the respiratory rate, the cECG was the most inaccurate in all stages, with AUC between 0.2 and 0.4 (cf. Figure 11). Coverage of only around 10% over all stages could be achieved. The Pearson correlation coefficients for these segment were above 0.8 (*p*-value < 0.05).

#### 5.2.2. Reflective PPG

The reflective PPG estimations of the heart rate with respect to the AUC were best for stages one and four. The best channel in this case achieved a median AUC of 0.6 and 0.7. For the stages with more motion, the AUC value dropped by around 0.1. In all cases, either ${\mathrm{rPPG}}_{3}$ or ${\mathrm{rPPG}}_{4}$ was the most accurate. In stage two, the reflective PPG was the least accurate with respect to all stages except for ${\mathrm{rPPG}}_{4}$. With the reflective PPG, median coverage of around 70% could be achieved. For these segments, Pearson correlation coefficients of above 0.75 were achieved (*p*-value < 0.05). With respect to the respiratory rate, the reflective PPG achieved median AUCs of around 0.5 for the stages with motion and around 0.6 for the motionless case. The performance was similar to that of the SCG for RR. Median coverage of around 25% for the respiratory rate could be achieved. The Pearson correlation coefficients in these cases were above 0.8 (*p*-value < 0.05).

#### 5.2.3. MIM

The MIM performed rather consistently between 0.4 and 0.5 with respect to the median AUC. All channels performed similarly. Median coverage of around 25% could be achieved with the MIM with respect to all stages. The Person correlation coefficients for these segments were above 0.81 (*p*-value < 0.05). In many cases, the MIM was disturbed by many spikes, which reduced the quality drastically.

#### 5.2.4. SCG

The SCG performed worst with respect to the AUC of the heart rate. Only in stage four, median AUCs of above 0.4 could be achieved. In the other stages, the median AUC was between 0.2 and 0.3. The median coverage was between 30% and 40% with Pearson correlation coefficients of above 0.76 (*p*-value < 0.05). For the respiratory rate, the SCG performed similarly to the reflective PPG in the first three stages, with median AUCs between 0.4 and 0.6 (cf. Figure 12). For stage four, the SCG outperformed the reflective PPG slightly, with median AUCs above 0.6. The SCG achieved coverage between 25% and 32%. The Pearson correlation coefficients were above 0.79 (*p*-value < 0.05).

With respect to the estimation of the RR, it should be noted that the minimum AUC and coverage across all channels and stages for each participant were at least 0.37 and 20% with medians of 0.59 and 34%, respectively. The maximum AUC and coverage achieved were 0.74 and 52%. For stage 4, the motionless case, the values were much higher, with a median AUC of 0.68 (minimum 0.47 and maximum 0.76) and median coverage of 45% (minimum 24% and maximum 64%).

In conclusion, it can be seen that cECG was the best modality for achieving low errors and high coverage with respect to the heart rate while also capturing HRV. The reflective PPG was second best in estimating HR and best in estimating RR with respect to the coverage including BRV. MIM and SCG did not achieve high coverage for RR but captured BRV well if the estimation was accurate enough. To achieve high AUCs and coverage for RR, the best channel or a fusion of several channels would be necessary. SCG was not well suited to capturing HR during movement.

## 6. Discussion

It was shown that the system could be used to extract the heart rate and respiratory rate from 20 healthy participants with coverage of at least 70% for the HR (according to IEC 60601-2-27) and 30% for the RR. However, improvements with respect to the hardware setup, study design and signal analysis are conceivable.

First, improvements of the hardware setup should be made with respect to the cECG setup and the MIM setup. We speculate that the cECGs’ signal quality can be further improved by reducing the cable length of the signal cables from the 4xU sensor to the controller box. In the current setup, the buffered cECG electrode potentials are connected to the ADS1298 by cables with a length of around 80 cm, which may introduce common-mode noise. With respect to the instrumentation, new active electrode setups should be investigated, which could include textile electrodes and improved electrode interfaces, which reduce the influence of movement. Specifically, the influence of long hair with respect to the discharge of triboelectricities needs to be investigated. Finally, the source of the 11.7 Hz disturbance should be further analysed and hardware changes should be performed accordingly. The disturbance probably arises from a mixture of two high-frequency signals, presumably from the voltage regulators. The MIM setup needs to be improved to reduce the spikes, which disturbed the signal. It was found that the spikes were generated by interruptions of the microcontroller during the measurement of the coil frequency due to the RS485 bus. Therefore, a measurement that cannot be interrupted should be implemented. Finally, a more reliable reference for the respiratory signal should be used to validate the estimated RR with respect to short-term variability. With respect to respiratory measurements, a respiratory belt could be helpful since it does not obstruct the face and is therefore less obtrusive than a mask.

Second, improvements with respect to the study population should be made. The study population should be more diverse with respect to ethnicity and gender and should include a larger number of participants. To analyse the effects of gender and hair length, the population should have a balance in gender and also include male participants with long hair. In addition, the age and BMI should be more diverse in such a way that also more older and heavier individuals participate. Since the weather can have an influence on the cECG signals, a study in varying environmental conditions (i.e., temperature and humidity) would be advantageous. Finally, the setup should be tested with patients suffering from cardiovascular diseases and in a real car.

Third, with respect to the signal analysis, sophisticated algorithms for sensor fusion should be investigated since all modalities and channels introduce redundancy, which could be used to further improve the results. Since some channels might provide a signal with higher quality than others, a selection according to a quality index is conceivable (cf., e.g., [48]). The cECG channels might also provide information beyond HR and HRV so that specific metrics to evaluate the signal shape for the cECG could be investigated. Furthermore, physiological parameters between modalities, such as pulse arrival times, should be investigated, especially since it is speculated that it may be usable as a surrogate for blood pressure [49]. These analyses could lead to new insights into whether the presented unobtrusive modalities are feasible for diagnostic applications in uncontrolled environments, e.g., at home during home care or for personal healthcare application with respect to the detection of unknown diseases of the cardiorespiratory system or monitoring of known diseases regarding a change in severity.

Since the cardiac-related SCG signals were mostly inaccurate, it is questionable whether this modality is useful for real-world driving scenarios, in which road and motor vibrations might further distort the signals. However, the accelerometer signals could still be used to detect motion artifacts. With this information, motion artifacts may be detected or even compensated. Further investigations in a real car are necessary. Since the cushion was designed as a portable device, it may also be usable in other private spaces, such as an armchair or even in office chairs (if the backrest provides proper mechanical contact). The disturbances in different scenarios with respect to movements should be investigated and compared since more movement can be expected in a vehicle than on a sofa at home. With respect to comfort, the cushion was not perceived as uncomfortable. However, a proper analysis for different private spaces is needed to verify this.

The provided analysis considers each modality independently and should give new insights into the problems and opportunities of each modality. Furthermore, implications for fusion algorithms as described above can be derived. The publicly available dataset may give further insights with respect to the quantification of motion artifacts and SNR in different driving situations and help to develop artifact detection and compensation techniques to make unobtrusive measurement feasible.

## 7. Conclusions

In this paper, a new, portable sensor cushion was introduced, which can be used to monitor the heart rate and respiratory rate of a vehicle driver. The recorded dataset was made publicly available in the UnoVis dataset. Furthermore, the cushion could be used in other private spaces. While the results are promising, improvements with respect to hardware and algorithms for vital sign extraction should be investigated.

## Figures and Tables

**Figure 1 sensors-23-04002-f001:**
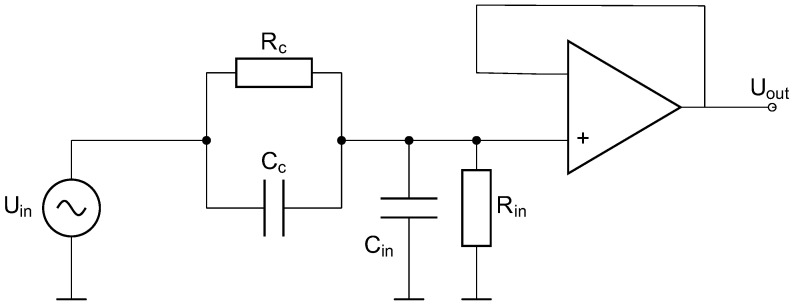
Model of active electrode (altered from [17]).

**Figure 2 sensors-23-04002-f002:**
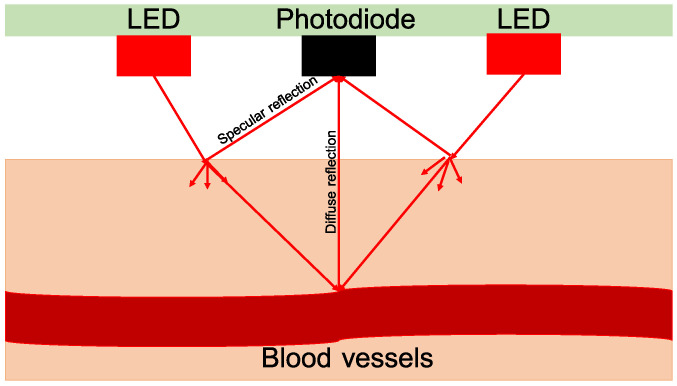
Principle of reflective PPG (inspired by [37]).

**Figure 3 sensors-23-04002-f003:**
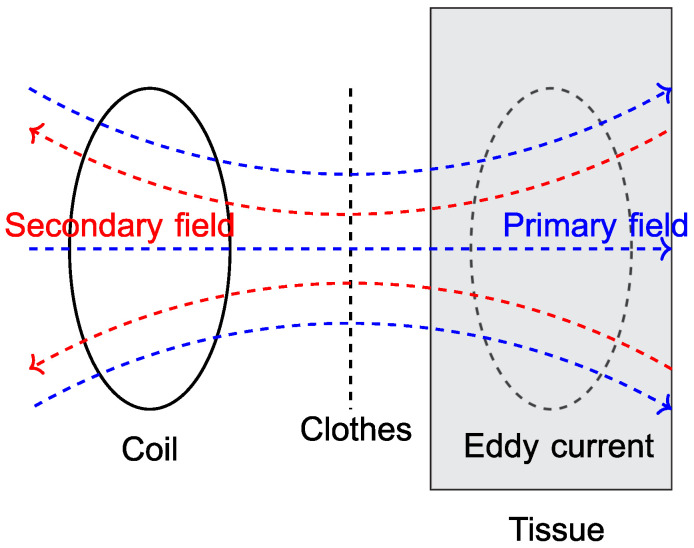
Principle of MIM (altered from [19]).

**Figure 4 sensors-23-04002-f004:**
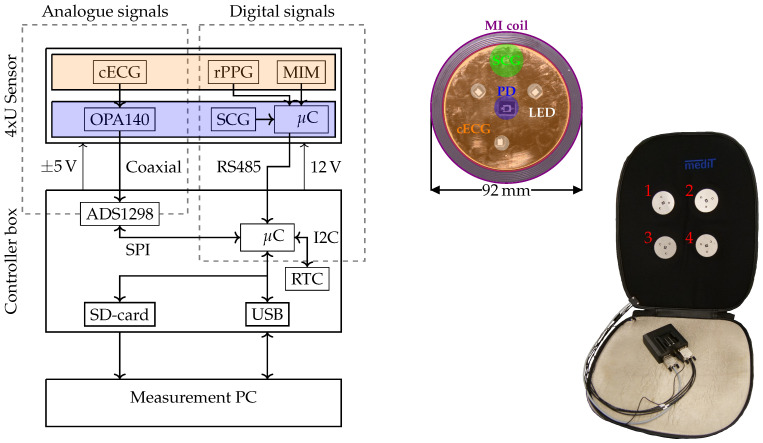
System overview. On the left side, a block diagram of the system is depicted. The orange highlighted block depicts the top side of the PCB (shown on the top right). The blue highlighted block depicts the bottom side of the PCB. On the right side, the cushion with the controller box is depicted.

**Figure 5 sensors-23-04002-f005:**
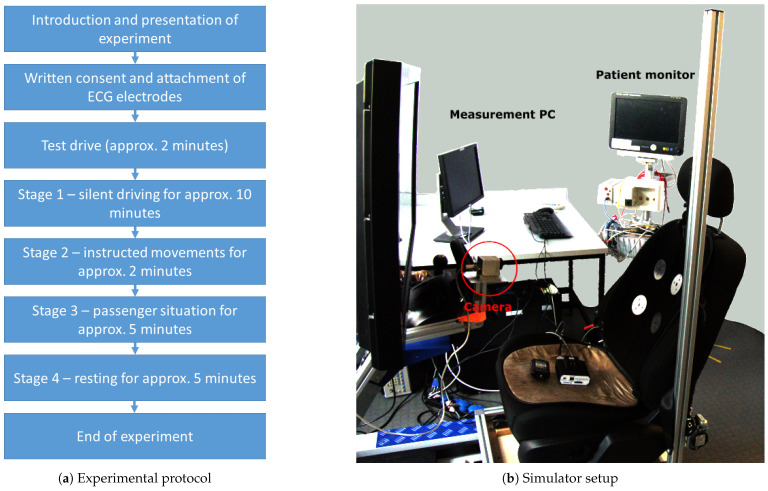
Simulator and protocol.

**Figure 6 sensors-23-04002-f006:**
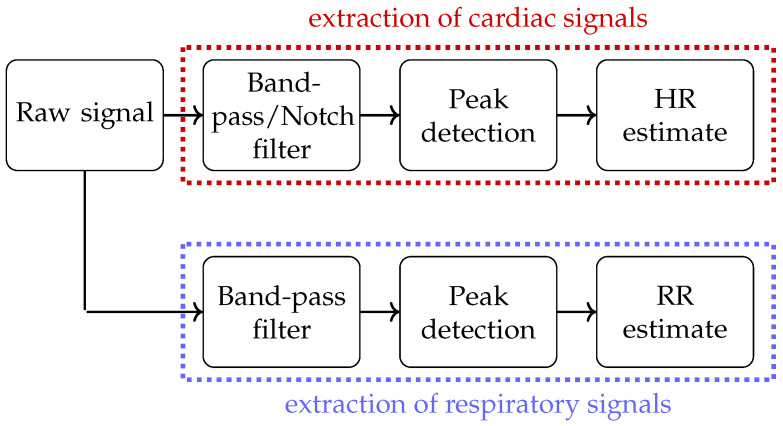
Workflow for processing of signals. Please note that the peak detection for the cECG signals is performed with the Pan–Tompkins algorithm and the algorithm of Brüser et al. is used to extract the HR of the SCG.

**Figure 7 sensors-23-04002-f007:**
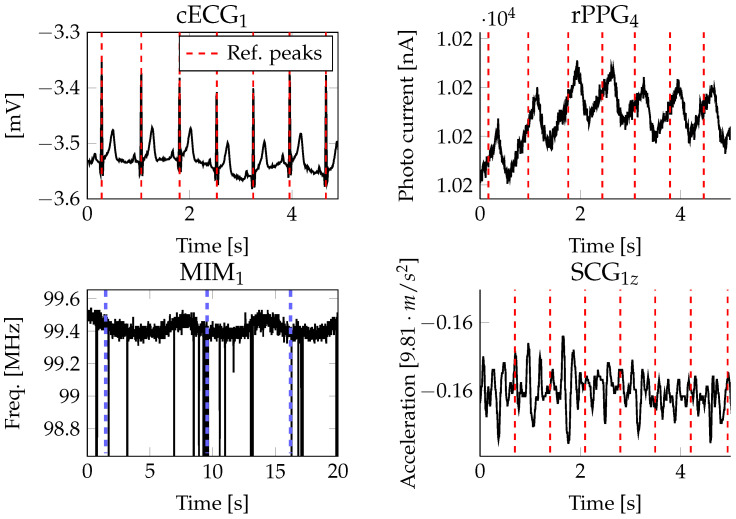
Signals with good quality. The reference peaks of the conductive ECG and impedance pneumography are shown as dashed red or blue lines, respectively.

**Figure 8 sensors-23-04002-f008:**
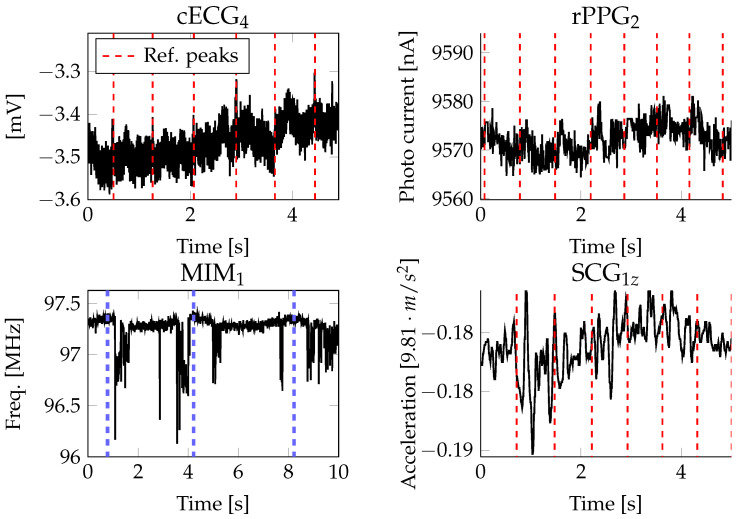
Signals with bad quality. The reference peaks of the conductive ECG and impedance pneumography are shown as dashed red or blue lines, respectively.

**Figure 9 sensors-23-04002-f009:**
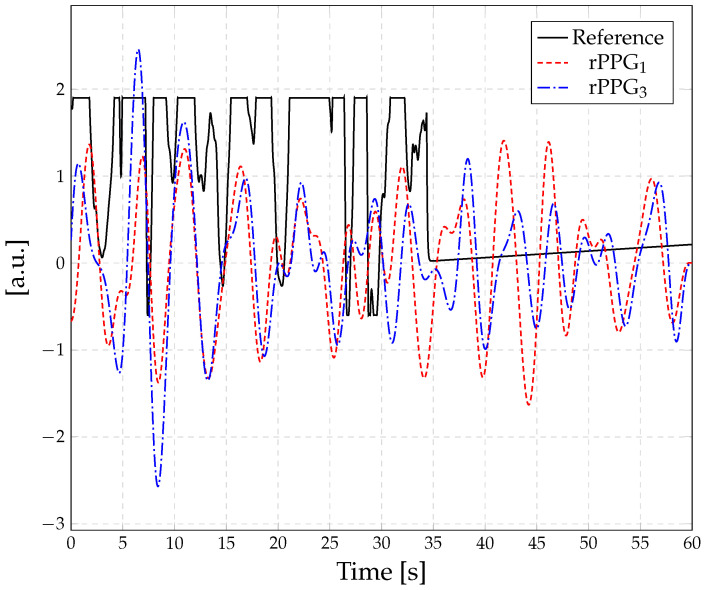
Example of unreliable respiratory reference. The respiratory signals extracted from rPPG${}_{1}$ and rPPG${}_{3}$ are shown for comparison.

**Figure 10 sensors-23-04002-f010:**
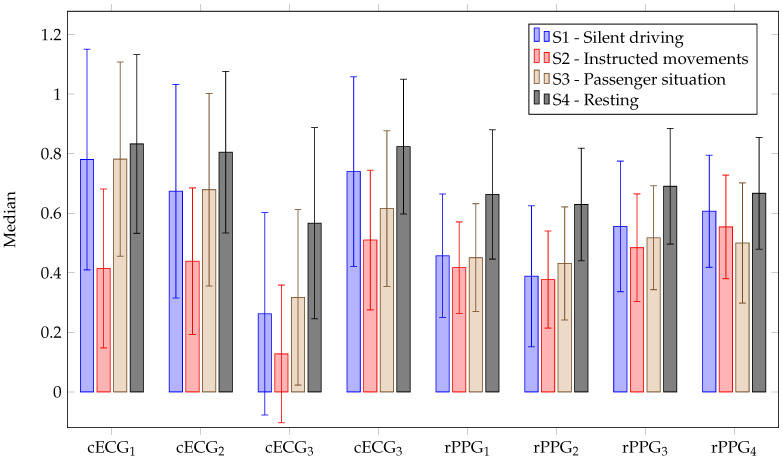
AUC of HR for different stages for cECG and reflective PPG. The bar shows the median across all participants and the lines show the standard deviation.

**Figure 11 sensors-23-04002-f011:**
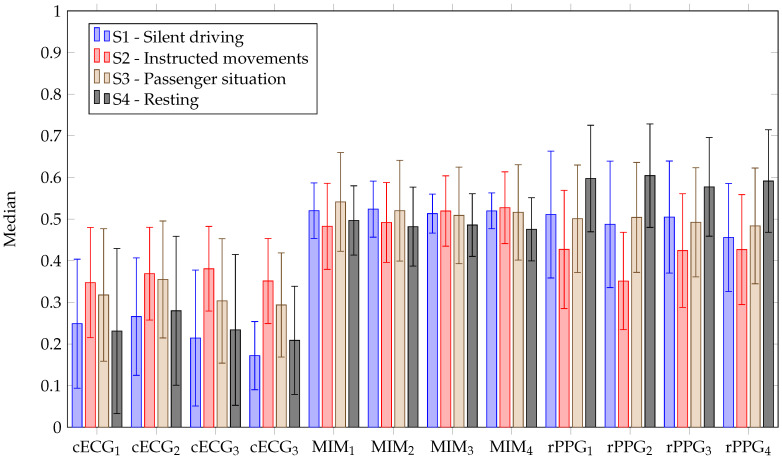
AUC of RR for different stages for cECG, MIM and reflective PPG. The bar shows the median across all participants and the lines show the standard deviation.

**Figure 12 sensors-23-04002-f012:**
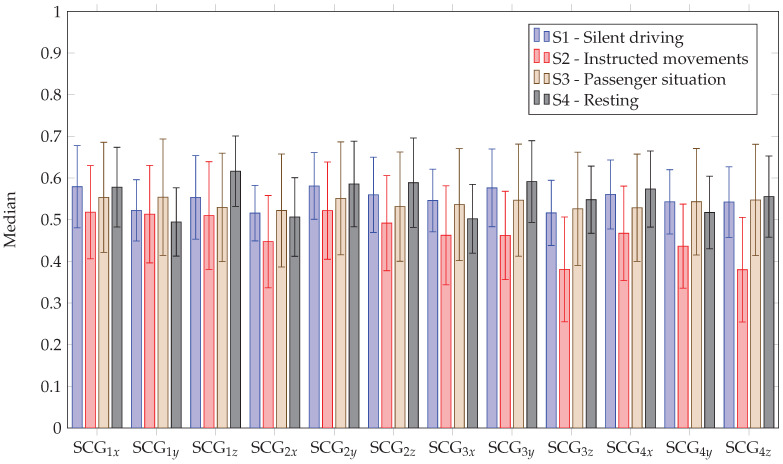
AUC of RR for different stages for SCG. The bar shows the median across all participants and the lines show the standard deviation.

**Table 1 sensors-23-04002-t001:** Table with demographics of participants, including clothes.

Participant	Age [Years]	Gender	Weight [kg]	Height [cm]	# Layers of Clothes	Material
1	24	male	70	174	1	cotton, polyester
2	22	male	70	186	1	cotton
3	21	male	92	203	1	cotton
4	20	male	80	181	1	cotton
5	19	male	60	165	2	cotton
6	24	female	61	165	1	cotton
7	25	male	75	184	1	cotton, polyester
8	22	male	93	187	1	cotton
9	27	male	80	180	1	cotton
10	26	female	66	170	1	cotton
11	23	male	71	187	1	cotton
12	23	male	80	183	1	cotton
13	28	male	82	173	1	cotton, polyester, spandex
14	23	male	78	174	1	cotton
15	25	male	93	180	1	cotton
16	29	male	77	189	1	cotton
17	22	male	71	180	1	cotton
18	25	male	65	178	1	cotton, polyester
19	30	male	80	174	1	cotton
20	60	male	65	172	2	cotton

**Table 2 sensors-23-04002-t002:** Coverage of HR with respect to segments fulfilling the requirements of IEC 60601-2-27 for each participant and all stages for cECG leads and reflective PPG channels.

Participant	${\mathrm{cECG}}_{1}$	${\mathrm{cECG}}_{2}$	${\mathrm{cECG}}_{3}$	${\mathrm{cECG}}_{4}$	${\mathrm{rPPG}}_{1}$	${\mathrm{rPPG}}_{2}$	${\mathrm{rPPG}}_{3}$	${\mathrm{rPPG}}_{4}$
1	0.852	0.893	0.517	0.883	0.695	0.538	0.614	0.666
2	0.023	0.104	0.025	0.374	0.649	0.383	0.766	0.736
3	0.959	0.954	0.958	0.935	0.597	0.384	0.374	0.395
4	0.754	0.749	0.345	0.724	0.875	0.879	0.92	0.832
5	0.928	0.938	0.826	0.945	0.855	0.721	0.84	0.837
6	0.3	0.135	0.21	0.705	0.186	0.758	0.391	0.712
7	0.793	0.781	0.692	0.83	0.784	0.713	0.538	0.689
8	0.74	0.737	0.71	0.682	0.737	0.711	0.729	0.767
9	0.872	0.789	0.882	0.761	0.853	0.889	0.866	0.954
10	0.037	0.018	0.021	0.061	0.334	0.39	0.821	0.705
11	0.829	0.876	0.733	0.794	0.568	0.608	0.736	0.814
12	0.901	0.858	0.26	0.855	0.741	0.599	0.712	0.739
13	0.048	0.165	0.027	0.248	0.868	0.921	0.779	0.837
14	0.918	0.572	0.339	0.556	0.868	0.868	0.788	0.862
15	0.793	0.805	0.779	0.811	0.573	0.436	0.391	0.59
16	0.852	0.917	0.331	0.878	0.614	0.471	0.707	0.672
17	0.893	0.912	0.69	0.852	0.876	0.8	0.918	0.932
18	0.568	0.741	0.264	0.694	0.173	0.17	0.174	0.17
19	0.821	0.512	0.163	0.382	0.884	0.858	0.876	0.93
20	0.266	0.219	0.148	0.371	0.501	0.269	0.74	0.866
median	0.8072	0.7650	0.3420	0.7424	0.7157	0.6590	0.7382	0.7530

## Data Availability

The data (not including camera data) were published in the UnoVis dataset (https://www.medit.hia.rwth-aachen.de/en/publications/unovis (accessed on 27 February 2023)). Camera data are available upon reasonable request.

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
