# Peer review of "A Portable Multi-Modal Cushion for Continuous Monitoring of a Driver’s Vital Signs"

_sensors, 2023, doi:10.3390/s23084002_

Round 1
Reviewer 1 Report
1. Could you please elaborate in the text on why the authors consider breathing rate and heart rate as the most crucial vital signs? This will provide more context and help readers understand the rationale behind this claim
2. It is worth noting that the abbreviation "rPPG" is commonly used to refer to remote photoplethysmography (PPG) using a camera in many papers. The authors have chosen to use "rPPG" to refer to reflective PPG in their paper. However, it may be more advisable to spell out "reflective photoplethysmography" in full throughout the text instead of using an abbreviation to ensure clarity and avoid reader confusion.
3. While the authors suggest that the driven right leg circuit is utilized to eliminate common mode noise in capacitive electrocardiogram (ECG) measurement, there appears to be insufficient explanation in the paper on how this circuit is implemented when the subject is wearing clothing. Additionally, the cited literature provided by the authors is a article that discusses the application of the driven right leg circuit to an ECG device that relies on direct skin contact. Further clarification is needed to fully understand the application of the driven right leg circuit in capacitive ECG measurement with clothed subjects
4. The paper does not provide a thorough explanation of how reflective photoplethysmography (PPG) can be measured accurately from subjects who are wearing clothing. While the authors mention that the illumination strength and LED wavelength must be appropriately set, this statement alone is insufficient to fully understand the technical implementation of this technique in clothed subjects. Furthermore, no related citation is provided to support this claim, making it difficult for readers to comprehend this content solely based on the information presented in this paper. Additional details and supporting references are needed to clarify the implementation of reflective PPG measurement in the presence of clothing.
5. The authors should provide further elaboration on the photograph presented in Figure 4. It may be advisable to modify the image to enable readers to identify the system configuration at a glance, including the location of the capacitive ECG electrode, LED, photodetector (PD), MIM coil, accelerometer, and other relevant components. This will enhance the reader's understanding of the experimental setup and facilitate interpretation of the results.
6. The authors should provide an explanation for why separate microcontrollers were used for the sensor unit and controller box. This would help readers understand the reasoning behind this design decision and the benefits that it provides. Possible reasons could include the need for dedicated processing power for each unit, isolation of functions to prevent interference or improve reliability, or ease of maintenance and upgradeability. Without additional information, it is difficult to determine the rationale behind the use of separate microcontrollers.
7. The authors should provide a more detailed explanation of the vital signs extraction process described in Section 4.2, specifically regarding the methodology used to measure respiratory rate. The current description of the signal processing techniques employed to extract the vital signs is insufficient and may not provide readers with a comprehensive understanding of the approach used. Additional information on how respiratory rate was measured from each signal, including specific signal processing techniques, would be helpful in clarifying the methodology. A more thorough explanation of the steps taken to extract each vital sign would enhance the reader's understanding of the results presented in this paper.
8. The definitions provided by the authors for "good" and "bad" signal quality of the measured vital signs are subjective and lack clarity. It is not clear how the authors established the criteria to determine whether signal quality was good, and how they reached consensus. To enhance the transparency and reproducibility of their results, the authors should provide a more detailed explanation of the methodology used to evaluate signal quality, including the objective metrics and criteria used to assess accuracy and precision. For example, the authors should provide detailed information on how they assessed the quality of the ECG signals, including how they determined that the P-wave, QRS complex, and ST-wave were accurately observed. By providing additional information on the criteria used to evaluate signal quality, the authors can ensure that other researchers can replicate their methodology and results, and promote a more comprehensive understanding of the limitations and potential biases of their approach.
9. The authors included several supplementary points in the discussion section, but some of these points should have been addressed in the current study, rather than deferred to future research. For example, the authors discussed the need to address issues related to cable length and hardware setup, and to use a more reliable comparator. However, these issues should have been fully addressed in the current study before presenting the results. Similarly, while the source of the 11.7 Hz noise can be investigated in future research, the authors should have included their current hypotheses regarding its cause in the paper. By providing a more thorough analysis of the data and addressing these issues in the current study, the authors can ensure that their results are as robust and reliable as possible, and contribute to a more comprehensive understanding of the limitations and potential biases of their approach.
10. The authors' opinion on the novelty of their paper is not clearly stated. The paper integrates four types of biometric technologies that have been previously published and demonstrates that they can be measured simultaneously. However, the authors should make a stronger claim regarding the technological progress achieved by proposing this integrated sensing technology. Specifically, the authors should highlight how the fusion analysis of multimodal signals presented in the third part of the discussion can lead to new insights and applications. By doing so, the authors can emphasize the significance of their work and its potential impact on the field of biometric sensing.
Reviewer 2 Report
The manuscript presents a biomedical signals' measuring system applicable to vehicles and other seating scenarios, aiming at estimating the heart rate and the respiratory rate of the subject. The manuscript is fairly well-written, but there are some typos that could be easily corrected by a thorough revision. The relevance of the manuscript is clear, but I believe the data needs more treatment before publication.
There are some smaller issues, as follows:
- the sentence "In partly autonomous vehicles, driver monitoring systems are crucial to ensure that the vehicle driver can take over control at any moment [4]." seems odd. In the case of a problem with the driver, shouldn't the VEHICLE take control?
- in line 70, the sentence "This from of ECG enables..." seems truncated;
- in line 78, it is a coupling CAPACITOR Cc, not a "conductor";
- in line 79, the correct term is "gigaohms", no caps;
The major issues that prevent the publication of the manuscript in its present form are:
- it is somewhat confusing which quantities each one of the four sensors are measuring. I understand there are 2 main biomedical quantities, HR and RR, and that some sensors can measure one, or both, but this information is diluted throughout the manuscript;
- in line 122, I understand that the system is composed by FOUR 4xU sensors, but this is not clear;
- the accelerometer sensor is not shown in Fig. 4;
- in line 145, HOW is the ambient light influence suppressed?
- in fig. 4, in the inset that shows one of the 4xU sensors, the individual sensors should be indicated;
- the choice of filters is odd. First, it is not clear if they are analog or digital filters (I believe the latter, but it is not explicit in the text). Then, there are some cascades of filters, as shown below, which make no sense regarding the various cutoff frequencies:
- cECG: BP 0.4 Hz-45 Hz ---> notch 50 Hz ---> BP 0.15 Hz - 0.5 Hz
- rPPG: BP 0.5 Hz - 2 Hz ---> BP 0.1 Hz - 0.4 Hz (the first BP would remove completely this band!)
- SCG: BP 0.4 Hz - 45 Hz ---> BP 0.1 Hz - 0.4 Hz (again)
- the graphs in Fig. 6 should all have the same time axis. Also, why arbitrary units? These quantities should have proper units. Finally, the red dashed lines in some case mean heart beats and in other cases respiratory cycles, which creates confusion;
- I did not understand the sentence "For the MIM, good quality means that the amplitude of the signal was in line with the respiration." in line 282;
- as a whole, the AUCs (especially for RR) seem very low, indicating that the system is still not usable in the practical sense. The authors should try some more postprocessing, including the suggested data fusion, in order to improve (a lot) the results before the manuscript can be published.
Round 2
Reviewer 1 Report
The authors diligently responded to the review comments and submitted a manuscript with many revised opinions reflected.
However, there seems to be a lack of new insights that readers can gain from this thesis other than the fact that four existing bio-signal measuring sensors have been made into one sensing module.
Reviewer 2 Report
The authors have properly addressed, corrected, or explained all the issues that I indicated in the previous revision, and I consider that the manuscript is now OK for publication, as it has been much improved. There are only two minor issues, as follows:
- in line 28, the term "repository" seems wrong. Shouldn't it be "respiratory"?
- in figure 8, the thousands separator (comma) should NOT be used, as it induces confusion (and it against the International Vocabulary of Metrology rules). Instead of "9,590", please use "9590" or "9 590".
